# Potential pandemic risk of circulating swine H1N2 influenza viruses

Valerie Le Sage[1,2], Nicole C. Rockey[1,9], Andrea J. French[1], Ryan McBride[3], Kevin R. McCarthy [1,2], Lora H. Rigatti[4], Meredith J. Shephard[5], Jennifer E. Jones [1], Sydney G. Walter[1], Joshua D. Doyle[2,6], Lingqing Xu[2,6], Dominique J. Barbeau [2,6], Shengyang Wang[3], Sheila A. Frizzell[7], Michael M. Myerburg[7], James C. Paulson [3], Anita K. McElroy [2,6], Tavis K. Anderson[8], Amy L. Vincent Baker[8] & Seema S. Lakdawala [1,2,5] ✉

Influenza A viruses in swine have considerable genetic diversity and continue to pose a pandemic threat to humans due to a potential lack of population level immunity. Here we describe a pipeline to characterize and triage influenza viruses for their pandemic risk and examine the pandemic potential of two widespread swine origin viruses. Our analysis reveals that a panel of human sera collected from healthy adults in 2020 has no cross-reactive neutralizing antibodies against a α-H1 clade strain (α-swH1N2) but do against a γ-H1 clade strain. The α-swH1N2 virus replicates efficiently in human airway cultures and exhibits phenotypic signatures similar to the human H1N1 pandemic strain from 2009 (H1N1pdm09). Furthermore, α-swH1N2 is capable of efficient airborne transmission to both naïve ferrets and ferrets with prior seasonal influenza immunity. Ferrets with H1N1pdm09 pre-existing immunity show reduced α-swH1N2 viral shedding and less severe disease signs. Despite this, H1N1pdm09-immune ferrets that became infected via the air can still onward transmit α-swH1N2 with an efficiency of 50%. These results indicate that this α-swH1N2 strain has a higher pandemic potential, but a moderate level of impact since there is reduced replication fitness and pathology in animals with prior immunity.

Influenza viruses cause acute respiratory infections in humans, and their wide host range provides many sources of strains with human pandemic potential. Influenza viruses exhibit strong host species preferences, which limits interspecies transmission, but they can evolve specific traits that allow sustained transmission within a new species. Although the major natural global reservoir of influenza virus is wild aquatic birds[1], swine are an important natural host and can act as a mixing vessel for reassortment of the eight viral gene segments of influenza A viruses from different host species. For example, the most recent H1N1 influenza virus pandemic from 2009 (H1N1pdm09) emerged from swine following reassortment events[2,3]. Emergence of future pandemic strains is a continuing threat

[1]Department of Microbiology and Molecular Genetics, University of Pittsburgh School of Medicine, Pittsburgh, PA, USA. [2]Center for Vaccine Research, University of Pittsburgh School of Medicine, Pittsburgh, PA, USA. [3]Departments of Molecular Medicine and Immunology & Microbiology, The Scripps Research Institute, La Jolla, CA, USA. [4]Division of Laboratory Animal Resources, University of Pittsburgh, Pittsburgh, PA, USA. [5]Department of Microbiology and Immunology, Emory University School of Medicine, Atlanta, GA, USA. [6]Division of Infectious Diseases, Department of Pediatrics, School of Medicine, University of Pittsburgh, Pittsburgh, PA, USA. [7]Department of Medicine, Division of Pulmonary, Allergy, and Critical Care Medicine, University of Pittsburgh School of Medicine, Pittsburgh, PA, USA. [8]Virus and Prion Research Unit, National Animal Disease Center, USDA-ARS Ames, IA, USA. [9]Present address: Department of Civil and Environmental Engineering, Duke University, Durham, NC, USA. ✉e-mail: seema.s.lakdawala@emory.edu

necessitating the monitoring and characterization of currently circulating swine viruses.

Influenza viruses are classified into subtypes based on the antigenicity of the surface viral glycoproteins, hemagglutinin (HA) and neuraminidase (NA). HA and NA are important determinants of virus infectivity, transmissibility, pathogenicity, and host specificity and evolve seasonally due to antigenic drift. In swine, three endemic subtypes predominate: swH1N1, swH1N2, and swH3N2 (Fig. 1A), which have roughly equal detections over the last four and a half seasons[4]. In the United States, the H1 classical swine lineage (1A) is divided into clades including α-H1 (1A.1), β-H1 (1A.2), and γ-H1 (1A.3), while the pre-2009 human seasonal-origin swine lineage (1B) includes the δ-H1 (1B.2) clades[5]. The majority of circulating swine strains distributed across the United States are classified within three genetically and antigenically distinct clades from the H1 1A classical swine lineage (1A.1.1.3, 1A.3.3.2, 1A.3.3.3: Fig. 1B, C and Supplementary Fig. 1)[4,6]. Since the 2010-2011 influenza season, there have been 18 H1N1, 35 H1N2 and 439 H3N2 infections in humans with variants of swine origin in the United States, with six from the α-H1 clade and 21 from γ-H1 clade (https://gis.cdc.gov/grasp/fluview/Novel_Influenza.html).

The current genetic diversity of influenza A virus (IAV) in swine reflects reassortment between avian-, swine-, and human-origin viruses, resulting in multiple lineages of the eight gene segments that continue to reassort among endemic swine strains. The subsequent antigenic drift of HA and NA while circulating in swine may result in viruses to which the human population may have little to no immunity[7]. Given the potential threat of such swine influenza viruses to humans, we created a decision tree to guide the characterization and pandemic risk assessment of endemic swine IAV (Fig. 2). Using a combination of both in vitro and in vivo methods, this decision tree capitalizes on the extensive research that has been conducted since the 2009 H1N1 pandemic on the molecular properties that promote efficient airborne transmission of influenza[8–20].

In this work, we assess the pandemic potential of the γ-H1 (1A.3.3.3) clade strain A/swine/Minnesota/A02245409/2020 (herein referred to as 'γ-swH1N1') and an α-H1 (1A.1.1.3) clade strain A/swine/Texas/A02245420/2020 (herein referred to as 'α-swH1N2') using this pipeline. These swine IAV clades were prioritized based on: detection frequency (Fig. 1B); geographical distribution (Fig. 1C); reported human variant events; significant loss in cross-reactivity to human seasonal vaccines or pre-pandemic candidate vaccine virus antisera[7]; limited detection by human population sera[7]; and interspecies transmission from pigs to ferrets[21]. We show that α-swH1N2 possesses more pandemic potential than γ-swH1N1 due to a lack of immunity in human serum samples, in vitro viral characteristics, and its ability to transmit via the air to both naïve ferrets and ferrets with prior immunity to seasonal influenza virus strains. Although α-swH1N1 can transmit between immune animals, its replication and disease severity appear to be reduced, suggesting that the impact of α-swH1N1 on human populations may be limited.

## Results

### Cross-reactivity of human sera against swine influenza viruses

A pandemic virus represents an antigenic shift, where a large proportion of the population is vulnerable due to a lack of immunity to this novel strain. To assess the presence of cross-reactive influenza virus-specific antibodies (Fig. 2, Box 1), human sera collected from healthy adults in Pennsylvania during the fall of 2020 were sorted by birth year and used in hemagglutination inhibition (HAI) (Fig. 3A) and neutralization assays (Fig. 3B). The prevalence of HAI and/or neutralizing antibodies against γ-swH1N1, α-swH1N2, or H1N1pdm09 was determined, and a threshold HAI titer of 40 was used as it is generally recognized as corresponding to a 50% reduction in the risk of infection[22,23]. H1N1pdm09 and γ-swH1N1-active antibodies as well as neutralizing antibodies were found across all birth year cohorts tested, whereas no HAI titer or neutralizing antibodies were detected against α-swH1N2 in any of the birth years tested (Fig. 3A, B). In addition, sera from individuals who recently received an influenza virus vaccine were tested to analyze samples with peak immunity from circulating antibodies (Fig. 3C). Recently vaccinated individuals had neutralizing

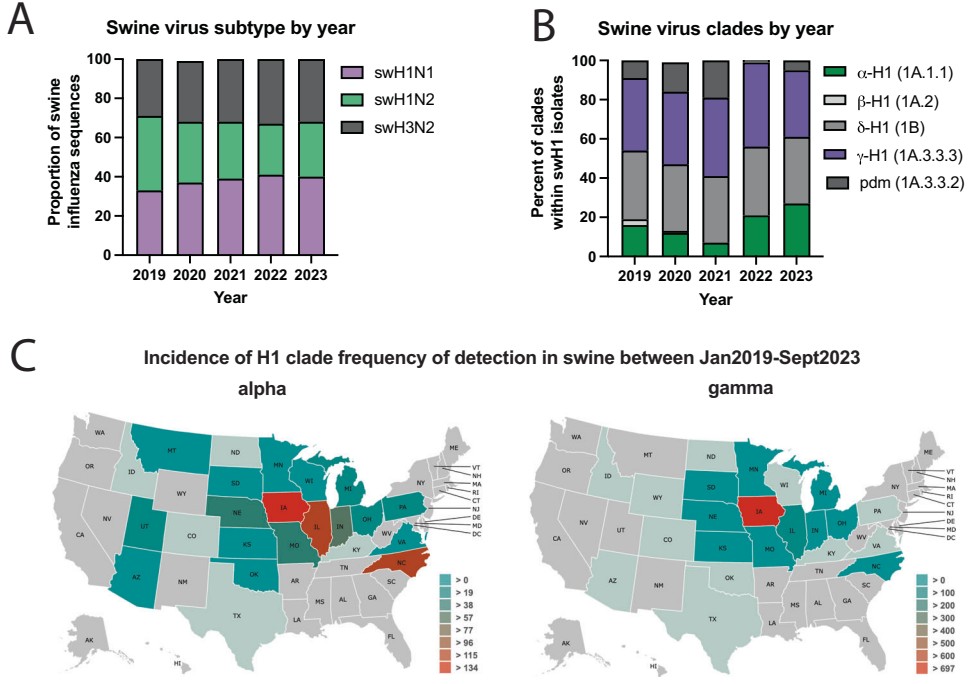

**Fig. 1 | Influenza A virus detected in swine between January 2019 and September 2023 in the USA. A** Influenza A virus subtype detection proportions. **B** H1 influenza A virus hemagglutinin clade detection proportions. pdm; pandemic. Data for A and B obtained from octoFLUshow[4]. **C** Detections of α-swH1N2 (alpha) and γ-swH1N1 (gamma) influenza A virus in swine across the United States between 2019 and 2021. Data retrieved with permission from ISU FLUture[65] on September 30, 2023.

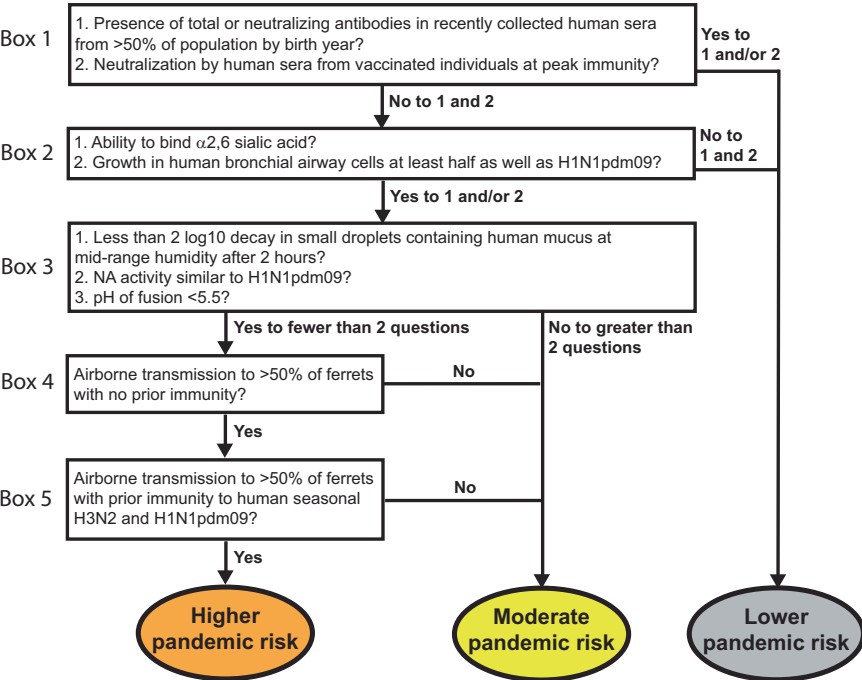

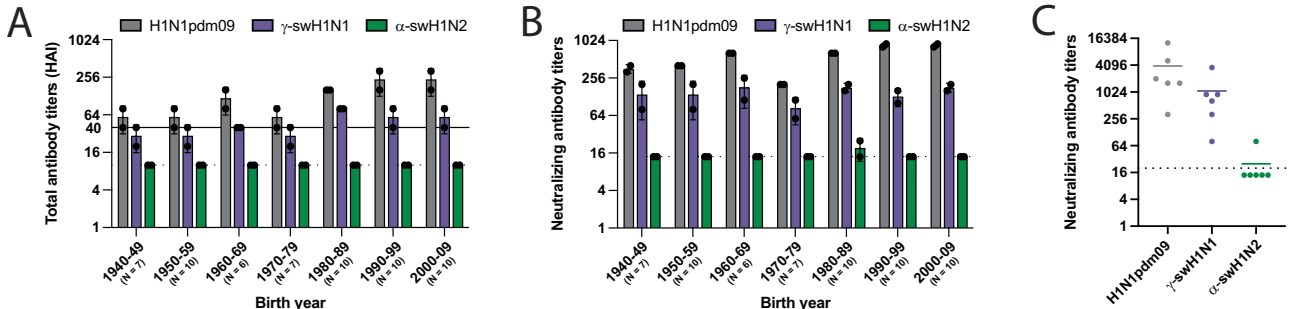

**Fig. 2 | Decision tree of influenza virus pandemic threat assessment.** Boxes include in vitro and in vivo methods to characterize influenza virus strains. Yes/no questions allow triage of strains into different pandemic risk assessment categories.

**Fig. 3 | Cross-reactivity of human sera to swine γ-H1N1 and α-H1N2 influenza viruses.** Pooled sera from the indicated number of humans for each decade of birth were tested for antibodies to H1N1pdm09 (red bars), γ-swH1N1 (purple bars), and α-swH1N2 (green bars) by HAI (**A**) and neutralization (**B**) assay. Data are presented as mean values +/− standard deviation (SD) from two biological replicates. Solid line in A indicates an HAI titer of 40, which corresponds to a 50% reduction in the risk of influenza virus infection. **C** Sera from individuals (N = 6) vaccinated in October 2021 (14 to 21 days post-vaccination) were assessed for cross-reactive neutralizing antibodies. Each dot represents an individual sample and is an average of 2 technical replicates. The colored lines represent the mean values between all the individual biological samples. For **A**–**C**, dashed lines indicate the limit of detection for each assay.

antibodies against H1N1pdm09 and γ-swH1N1, but not α-swH1N2 (Fig. 3C). Based on the decision tree (Fig. 2, Box 1), the presence of cross-reacting antibodies against γ-swH1N1 would funnel the virus to a lower pandemic risk, while α-swH1N2 would require further characterization. However, for this study, we proceeded to characterize both γ-swH1N1 and α-swH1N2 to provide empirical evidence for the decision tree criteria.

**Molecular characterization of swine strains**
The H1N1pdm09 HA segment is of swine-origin from the classical H1 lineage[3]. To examine the similarities between the three strains, amino acid differences of the γ-swH1N1 (Supplementary Fig. 2A) and the α-swH1N2 (Supplementary Fig. 2C) HA were mapped onto the H1N1pdm09 HA structure. The γ-swH1N1 HA has 46 amino acid differences as compared to the H1N1pdm09 HA, while α-swH1N2 has 86. Similarity between γ-swH1N1 and H1N1pdm09 HA likely accounts for the cross-neutralizing and cross-receptor blocking antibodies present

in human serum (Fig. 3 and Supplementary Fig. 2A, purple residues). Diversity in the α-swH1N2 HA is greatest in the globular HA head domain, at sites surrounding the receptor binding site (RBS) (130-strand, 140-loop, 150-loop, 190-helix and the 220-loop[24]) (Supplementary Fig. 2C, green residues). The otherwise conserved RBS is responsible for engaging cell surface sialic acids (SA). In the 130-strand, α-swH1N2 has a two-amino acid deletion (Supplementary Fig. 2D, yellow residues), which may impact antibody binding. Additionally, γ-swH1N1 and α-swH1N2 have an additional putative glycosylation site at the same position on the side of the HA head domain, whereas α-swH1N2 has a second putative glycosylation site near the apex of the HA and its three-fold axis of symmetry (Supplementary Fig. 2A, C, pink residues). The evolution of glycosylation sites is thought to contribute to immune escape by shielding antigenic sites on HA[25–27]. During the H1N1 2009 pandemic, differences in the number of putative glycosylation sites between H1N1pdm09 and seasonal viruses were associated with the lack of cross-neutralizing antibodies[28]. Differences in amino

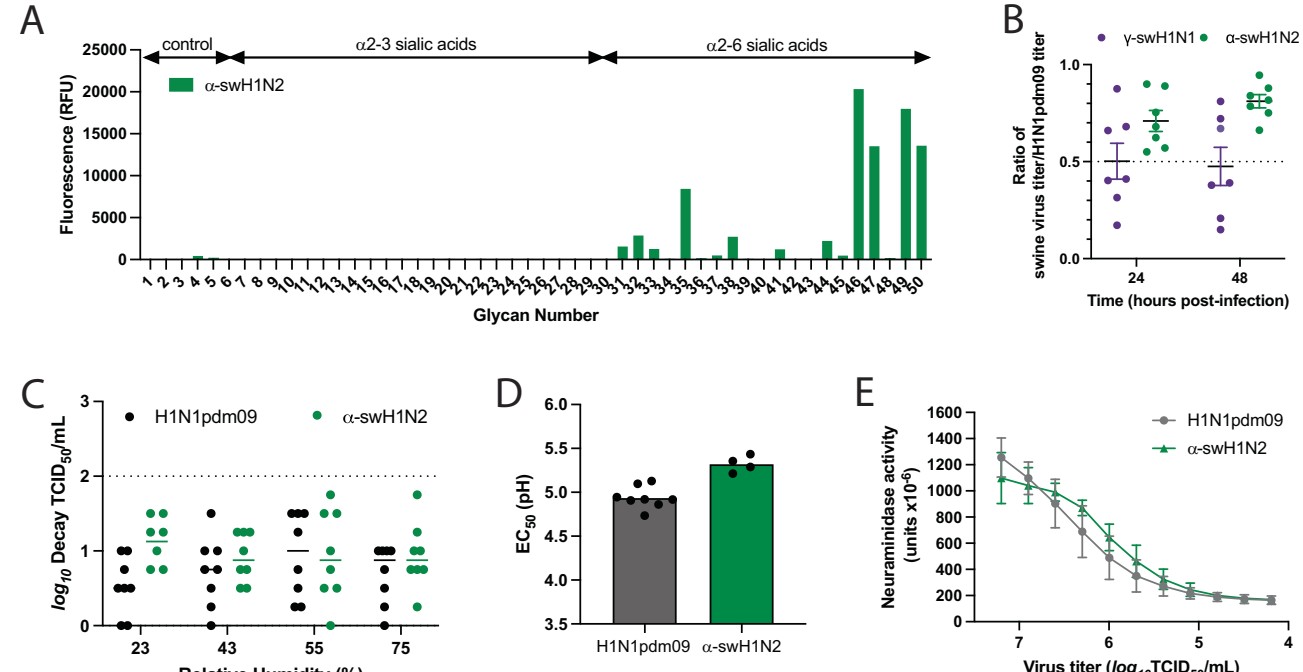

**Fig. 4 | In vitro characterization of swine γ-H1N1 and α-H1N2 influenza viruses.**
**A** Binding of α-swH1N2 virus to a sialoside microarray containing glycans with α2-3 or α2-6 linked sialic acids representing avian-type and human-type influenza receptors, respectively. Bars represent the fluorescence intensity of bound α-swH1N2. Glycan structures corresponding to numbers are shown on the x-axis are found in Supplementary Table 1. Signal values are calculated from the mean intensities of 4 of 6 replicate spots with the highest and lowest signal omitted.
**B** Replication of swine influenza virus in human bronchial epithelial (HBE) air-liquid interface cell cultures. HBE cell cultures were infected in triplicate with $10^3$ TCID$_{50}$ (tissue culture infectious dose 50) per well of H1N1pdm09, γ-swH1N1, or α-swH1N2. The apical supernatant was collected at the indicated time points and virus titers were determined on MDCK cells using TCID$_{50}$ assays. A ratio of swine virus titer relative to H1N1pdm09 titer at 24 and 48 h of all HBE patient cell cultures is shown. Each dot represents an average of three technical replicates per HBE culture, and seven biological replicates from different HBE patient cultures are displayed. Data are presented as mean values +/− SD of the seven biological replicates each with

three technical replicates. **C** Stability of α-swH1N2 influenza virus in small droplets over a range of relative humidity (RH) conditions. Ten 1 uL droplets of pooled virus from panel B were spotted into the wells of a tissue culture dish for 2 h. Decay of the virus at each RH was calculated compared to the titer of ten 1 uL droplets deposited and immediately recovered from a tissue culture dish. Log$_{10}$ decay of HBE-propagated H1N1pdm09 (black) and α-swH1N2 (green) is shown and represents mean values (±SD) from eight biological replicates performed in three technical replicates. **D** H1N1pdm09 (gray, $N = 8$) and α-swH1N2 (green, $N = 4$) viruses were incubated in PBS solutions of different pHs for 1 h at 37 °C. Virus titers were determined by TCID$_{50}$ assay and the EC$_{50}$ values were plotted using regression analysis of the dose-response curve. The reported mean corresponds to at least four independent biological replicates, each performed in three technical replicates. **E** The NA activities of H1N1pdm09 (gray) and α-swH1N2 (green) were determined using an enzyme-linked lectin assay with fetuin as the substrate. Viruses were normalized for equal infectivity and displayed data are the mean (±SD) of three independent biological replicates performed in technical duplicates.

acids and glycosylation sites in the α-swH1N2 HA head could contribute to the lack of detectable cross-reactive antibodies observed in Fig. 3 compared to the γ-swH1N1 or alter receptor preference.

Receptor preference of influenza A viruses is a critical host adaptive property and one known to be important for successful adaptation of influenza viruses to the human population[29]. Human and swine influenza viruses are known to have an α2-6 SA preference, while avian influenza viruses have an α2-3 SA preference. Analysis of H1N1pdm09 and recently circulating human seasonal H3N2 viruses suggests that human viruses adapt to preferential recognition of extended glycans capped with α2-6 SA[29–33]. Analysis of α-swH1N2 receptor specificity using a glycan array with a focused panel of α2-3- and α2-6-linked sialoside glycans showed a strict specificity for glycans with α2-6 sialic acids. For N-linked glycans extended with 1-3 LacNAc (Galβ1-4GlcNAc) repeats, clear preference is shown for extended glycans with two (#46, #49) or three (#47, #50) LacNAc repeats over those with a single LacNAc repeat (#45, #48) (Fig. 4A and Supplementary Table 1). Thus, the α-swH1N2 virus exhibits a receptor specificity well adapted for human-type receptors.

To assess fitness of swine viruses to replicate within the human respiratory tract, replication capacity of γ-swH1N1 and α-swH1N2 was determined in human bronchial epithelial (HBE) patient cell cultures grown at an air-liquid interface (Fig. 4B). Multiple human HBE cultures

were tested, and an H1N1pdm09 virus control was included in all experiments. The ratio of swine virus titer over H1N1pdm09 virus titer for each HBE culture is reported. The representative γ-swH1N1 strain replicated approximately half as well as H1N1pdm09, whereas the representative α-swH1N2 strain had a titer ratio of 0.71 and 0.81 at 24 and 48 h, respectively (Fig. 4B). These data indicate that, regardless of deletions in the RBS 130-loop (Supplementary Fig. 2D, yellow residues), α-swH1N2 replicates to levels similar to H1N1pdm09 (Fig. 2, Box 2) and would support α-swH1N2 being selected for additional characterization of parameters correlated with efficient human-to-human transmission of influenza viruses (Fig. 2, Box 3).

Airborne transmission requires viral persistence in expelled aerosols and droplets, which can be influenced by environmental conditions, including relative humidity (RH)[34] or respiratory secretions like HBE airway surface liquid[35,36]. To study the impact of RH on influenza virus viability, droplets of H1N1pdm09 and α-swH1N2 viruses propagated from HBE cultures in Fig. 4B were exposed to different RH conditions. HBE-propagated H1N1pdm09 and α-swH1N2 experienced very little decay in infectivity at all RH tested (Fig. 4C). These data indicate that α-swH1N2 expelled in small droplets in the presence of human respiratory secretions remains viable over a range of RH conditions, which is important for efficient airborne transmission and viral persistence.

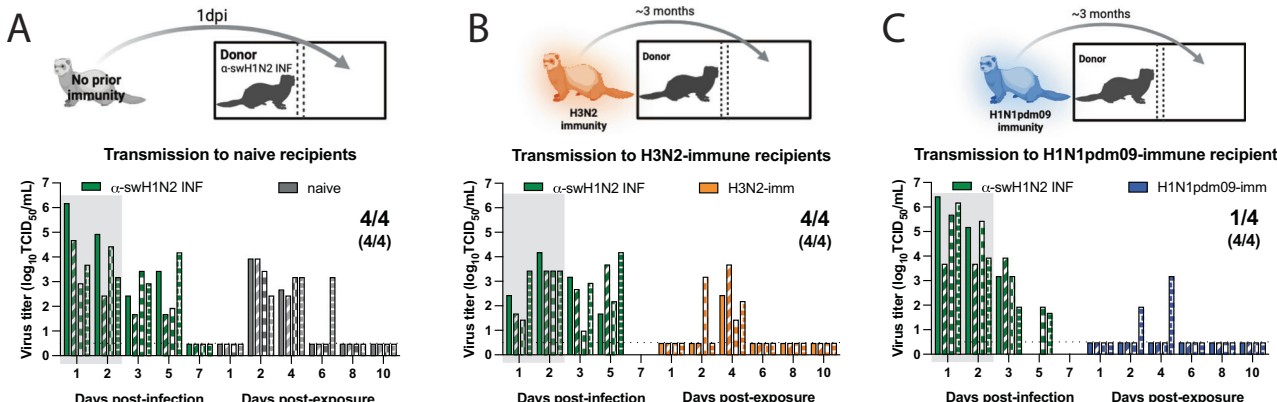

**Fig. 5 | Swine α-H1N2 transmits efficiently via the air after a short exposure.**
**A** Schematic of experimental procedure to naïve recipients. Shaded gray box depicts exposure period. Four donor ferrets were infected intranasally with α-swH1N2 (α-swH1N2 INF), as in Methods. Recipient ferrets with no prior immunity (naïve recipients) were placed in the adjacent cages at 24 h post-infection for two continuous days. **B** Schematic of procedure, whereby four ferrets were infected with H3N2 A/Perth/16/2009 strain (H3N2-imm) 137 days prior to acting as recipients to α-swH1N2 infected donors. Four donor ferrets were infected with α-swH1N2 and H3N2-imm recipients were placed in the adjacent cage 24 h later. **C** Schematic of procedure, whereby four ferrets were infected with H1N1pdm09 (H1N1pdm09-imm) 126 days prior to acting as recipients to α-swH1N2 infected

donors. H1N1pdm09-imm recipients were placed in the adjacent cage 24 h later. Nasal washes were collected from all ferrets on the indicated days and titered for virus by TCID50 (tissue culture infectious dose 50). Each bar indicates an individual ferret. Pairs of ferrets are matched with the shading type within the bar. For all graphs, the number of recipient ferrets with detectable virus in nasal secretions out of four total is shown; the number of recipient animals that seroconverted at 14- or 21-days post α-swH1N2 exposure out of four total is shown in parentheses. Gray shaded box indicates shedding of the donor during the exposure period. The limit of detection is indicated by the dashed line. Schematics in **A–C** were created with BioRender.com.

In addition to receptor binding, HA-mediated membrane fusion between the viral envelope and cellular endosome is required for viral entry and is driven by pH changes. A conformational change in the HA from human influenza viruses is triggered between pH 5.3 and 5.5, while avian HA proteins are triggered at a higher pH range of 5.5 to 6.2, suggesting that human adaptation necessitates increased acid stability[37]. To determine the pH at which HA undergoes its conformational change, an acid stability assay was performed on H1N1pdm09 and α-swH1N2, as a surrogate for the pH of fusion[20,38]. The pH that reduces the viral titer by 50% ($EC_{50}$) for α-swH1N2 was 5.3, which was similar to H1N1pdm09 at 5.0 (Fig. 4D), indicating that α-swH1N2 has a pH of fusion comparable to human influenza viruses, which is below pH 5.5.

The neuraminidase activity of the NA receptor is necessary to cleave SA from the host cell surface and release the virus. A functional balance between HA and NA is necessary for airborne transmission of swine viruses[18,19,39]. Higher NA activity has also been implicated in the efficient airborne transmission of H1N1pdm09 compared to its swine precursor strains, which had very little NA activity[12]. To measure NA activity, we used an enzyme-linked lectin assay with fetuin as a substrate and a bacterial neuraminidase standard. The NA activity of α-swH1N2 was observed to be similar to that of H1N1pdm09 (Fig. 4E). Taken together, these in vitro results indicate that α-swH1N2 has the molecular features consistent with a virus capable of airborne transmission and requires further characterization.

## Swine α-H1N2 airborne transmission in ferrets

Following the decision tree criteria (Fig. 2), we next characterized α-swH1N2 in vivo for the efficiency of airborne transmission in the ferret model (Fig. 2, Box 4). Epidemiologically successful human seasonal influenza viruses transmit to naïve recipients after a 2-day exposure[40]. Using this methodology, experimentally infected α-swH1N2 donors were housed with naïve recipients in cages where the animals were separated by a divider. A successful transmission event was defined as recovery of infectious virus in recipient nasal secretions or seroconversion at 21 days post-infection (dpi). In the infected donors, α-swH1N2 was detected in nasal secretions on 1, 2, 3 and 5 dpi (Fig. 5A, green bars). Four of four recipients without prior immunity shed α-swH1N2 starting 2 days post-exposure (dpe) (Fig. 5A, gray bars). All

recipient animals seroconverted at 14 dpi, with increases in antibody titers the following week (Supplementary Table 2). These data indicate that α-swH1N2 transmits efficiently to animals without prior immunity within a short 2-day exposure, similar to published reports of H1N1pdm09[40].

Pandemic influenza viruses do not emerge in immunologically naïve populations as most individuals have experienced influenza by the age of 5[41]. We have previously established a pre-immune ferret model that can be used to assess the pandemic potential of emerging strains in the context of prior immunity[40]. To determine the impact of pre-existing immunity on the transmission efficiency of α-swH1N2, four recipient ferrets were first infected with the H3N2 A/Perth/16/2009 strain ('H3N2-imm recipient') or H1N1pdm09 ('H1N1pdm09-imm recipient'). Roughly 4 months later, once the response to the primary infection was allowed to wane[40,42–45], these ferrets were then exposed to infected α-swH1N2 donors for 2 days (Fig. 5B and C). In replicate 1, four of four H3N2-imm recipients shed α-swH1N2 at 4 dpe (Fig. 5B), whereas only two of four shed virus in replicate 2 (Supplementary Fig. 3). All H3N2-imm recipients that shed virus also seroconverted with increasing antibody titers over time (Supplementary Table 2). All four of four H1N1pdm09-imm recipients seroconverted at 13dpe and had rising antibody titers at 20 dpe (Supplementary Table 2). Intriguingly, only one of four H1N1pdm09-imm recipients shed detectable levels of α-swH1N2 (Fig. 5C). It is possible that shedding of virus was missed in the recipients either because the nasal wash samples were not taken at 3 dpe or that replication of the virus was occurring in a place that was not sampled by the nasal wash, such as the mid-turbinates, nasopharynx, trachea, or lungs. However, based on serology we can conclude that all four H1N1pdm09 pre-immune animals were infected (Supplementary Table 2). These data suggest that α-swH1N2 can transmit to animals with prior immunity, which categorizes α-swH1N2 into the higher pandemic risk. However, whether naturally infected ferrets with prior immunity could spread the virus onward is unclear.

## Potential α-swH1N2 pandemic severity

Person-to-person airborne transmission is a concern for pandemic emergence and can be experimentally assessed using transmission

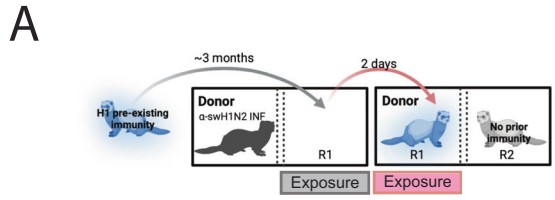

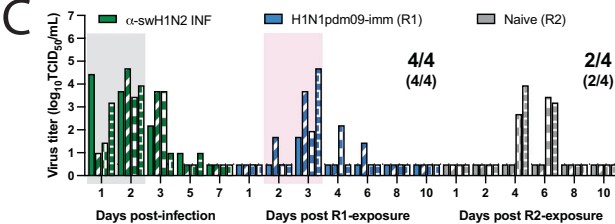

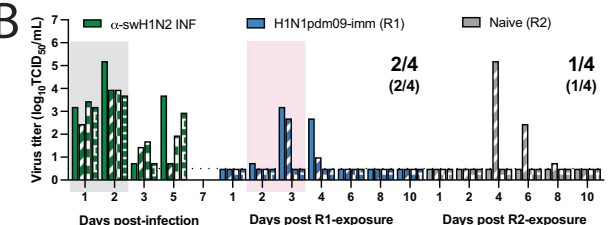

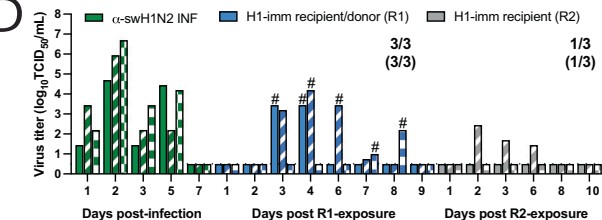

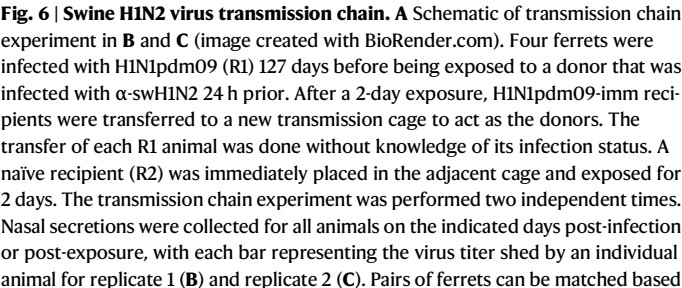

**Fig. 6 | Swine H1N2 virus transmission chain. A** Schematic of transmission chain experiment in **B** and **C** (image created with BioRender.com). Four ferrets were infected with H1N1pdm09 (R1) 127 days before being exposed to a donor that was infected with α-swH1N2 24 h prior. After a 2-day exposure, H1N1pdm09-imm recipients were transferred to a new transmission cage to act as the donors. The transfer of each R1 animal was done without knowledge of its infection status. A naïve recipient (R2) was immediately placed in the adjacent cage and exposed for 2 days. The transmission chain experiment was performed two independent times. Nasal secretions were collected for all animals on the indicated days post-infection or post-exposure, with each bar representing the virus titer shed by an individual animal for replicate 1 (**B**) and replicate 2 (**C**). Pairs of ferrets can be matched based

on the patterns in the bars. Gray shaded boxes indicate the days upon which the α-swH1N2 infected (α-swH1N2 INF) donor was exposing the H1N1pdm09-imm recipient (R1), and the pink shaded box indicates the days upon which R1 was acting as the donor to R2. Limit of detection is denoted by a dashed line. The numbers in parentheses indicate the proportion of animals that seroconverted. **D** Donors were infected with α-swH1N2 24 h prior to exposing H1N1pdm09-imm recipients. Nasal washes from R1 recipients were collected and immediately tested using a rapid antigen test. Once positive for influenza virus antigen, the H1N1pdm09-imm R1 recipient was moved into a new transmission cage to act as the donor and expose an H1N1pdm09-imm R2 recipient for 2 days. # indicates the 2-day window in which each of the R1 ferret exposed the R2 recipients.

chain experiments. Given the lack of detectable shedding of α-swH1N2 in three of four H1N1pdm09-imm recipients yet seroconversion in all four recipients in Fig. 5C, we examined whether H1N1pdm09-imm recipients would shed enough virus to onward transmit α-swH1N2 to naïve recipients. Two independent replicate transmission chains were performed with four α-swH1N2 infected donors being exposed to H1N1pdm09-imm recipients in the adjacent cage for 2 days. The exposed H1N1pdm09-imm recipients (R1) were then transferred to a new cage to act as donors to naïve recipients (R2) (Fig. 6A). In the first replicate (Fig. 6B), two of four R1 ferrets shed α-swH1N2, whereas in the second replicate (Fig. 6C), all four R1 ferrets had α-swH1N2 in their nasal secretions. Much of the viral shedding was observed on day 3 post exposure, which may account for the absence of robust shedding in Fig. 5C. When R1 ferrets became donors (Fig. 6B, C, pink box), only 50% of the infected donors transmitted α-swH1N2 onward to influenza immunologically naïve recipients. To further examine the ability of α-swH1N2 to transmit from H1N1pdm09-imm ferrets in the context of pre-existing immunity, a transmission chain experiment was performed using R2 recipients with H1N1pdm09 immunity (Fig. 6D). To ensure that the exposure window of viral shedding of R1 was captured, rapid antigen tests were performed immediately following the nasal wash; when positive, that animal was transferred into a cage to act as a donor animal to an H1N1pdm09 R2 immune ferret (Fig. 6D). In this study, all three R1 became infected and of these three α-swH1N2-infected H1N1pdm09-imm R1 recipients only one onward transmitted to the H1N1pdm09-imm R2 recipients (Fig. 6D). Seroconversion only occurred in R1 and R2 recipients that shed detectable α-swH1N2 virus (Supplementary Table 2). These data suggest that onward transmission of α-swH1N2 is possible, even in the context of pre-existing immunity, contributing to a higher risk potential of α-swH1N2.

We next examined the impact of prior influenza virus exposure on α-swH1N2 replication and pathogenesis; ferrets with no prior immunity, pre-existing immunity against H3N2 (H3N2-imm) or H1N1pdm09 (H1N1pdm09-imm) were intranasally infected with α-swH1N2 and their

nasal secretions were collected over time. No difference in α-swH1N2 titers was observed between H3N2-imm ferrets and those with no prior immunity from 1-3 dpi, however, H3N2-imm ferrets cleared α-swH1N2 by 5 dpi (Fig. 7A). This observation is consistent with our previous reports of a reduced viral shedding period in animals with heterosubtypic immunity[40,46]. Interestingly, ferrets with pre-existing H1N1pdm09-imm shed significantly less α-swH1N2 virus on 1, 2, and 3 dpi as compared to ferrets with no prior immunity (Fig. 7B).

To characterize tissue-specific α-swH1N2 replication, infected ferrets from panels 7A and 7B were sacrificed at 5 dpi and the respiratory tract was collected for viral titration (Fig. 7C). In ferrets with no prior immunity, robust replication was detected in the lungs, trachea, soft palate, and nasal turbinates, whereas H3N2-imm ferrets only had detectable α-swH1N2 in the soft palate (Fig. 7C). H1N1pdm09-imm ferrets had completely cleared α-swH1N2 from their respiratory tracts on day 5, as no detectable infectious virus was detected in any of the collected tissues (Fig. 7C). Since viral titers from nasal washes were already reduced in these animals by 3 dpi, we next assessed viral replication in the respiratory tract at this time point. H1N1pdm09-imm and non-immune ferrets were infected with α-swH1N2 and sacrificed on day 3 (Fig. 7D). H1N1pdm09-imm ferrets had detectable α-swH1N2 in the respiratory tract, although viral titers were significantly less in the lungs and nasal turbinates compared to animals without prior immunity. Taken together, these data indicate that prior H1N1pdm09 immunity can reduce the viral load in the ferret respiratory tract and decrease time to clearance of α-swH1N2.

To extend the observation on viral titers, we compared the lung pathology of α-swH1N2-infected ferrets with no prior immunity to those with H1N1pdm09 pre-existing immunity (from Fig. 7D). Regardless of immunity, at 3 dpi α-swH1N2-infected ferrets had bronchial glands that were multifocally necrotic and contained neutrophils as well as peripheral lymphocytes, whereas uninfected ferrets had intact glands and no inflammation (Supplementary Fig. 4). The bronchioles from infected ferrets with no prior immunity were

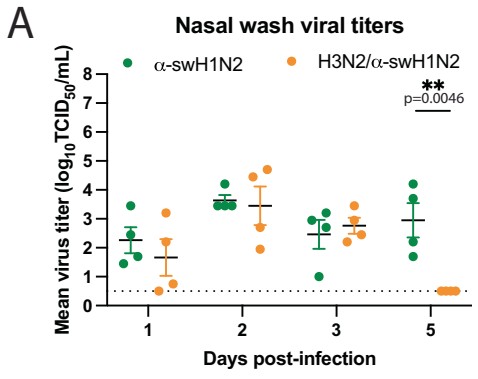

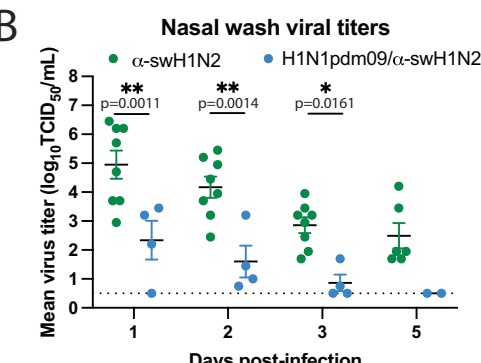

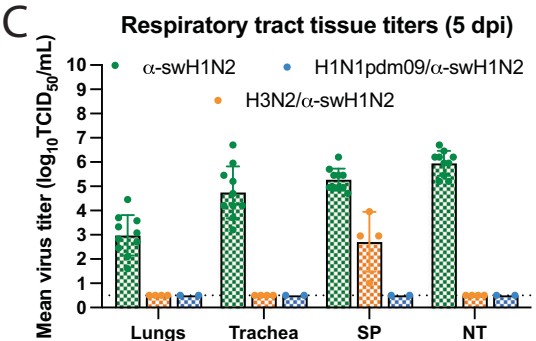

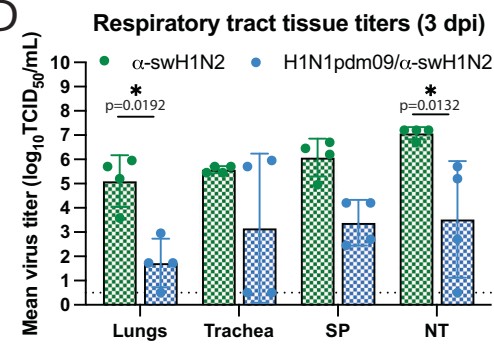

**Fig. 7 | H1N1pdm09-immune ferrets have reduced α-swH1N2 viral titers in nasal secretions and tissues. A** Ferrets with no pre-existing immunity (α-swH1N2, $N = 4$) or those infected with H3N2 137 days prior (H3N2/α-swH1N2, $N = 4$) were intranasally infected with α-swH1N2. The mean ± SD viral titers from nasal secretions are shown with each circle representing an individual animal. Two-way ANOVA analysis was used to determine statistically significant differences. **B** α-swH1N2 mean ± SD viral titers from nasal secretions from animals with no prior immunity (α-swH1N2, $N = 8$) or those infected with H1N1pdm09 126 days prior (H1N1pdm09/α-swH1N2, $N = 4$). Two-way ANOVA analysis was used to determine statistically significant differences. **C** Respiratory tissues from α-swH1N2 infected ferrets with no prior

immunity (green; $N = 10$), H3N2 prior immunity (orange; $N = 4$), or H1N1pdm09 prior immunity (blue; $N = 2$) were collected at 5 dpi. Graphs show the mean hatched bar ± SD of viral titers for all biological replicates presented as individual data points. SP-soft palate, NT-nasal turbinates. (**D**) Respiratory tissues from α-swH1N2 infected ferrets with no prior immunity (green; $N = 4$) or H1N1pdm09 prior immunity (blue; $N = 4$) were collected at 3 dpi. Graphs show the mean hatched bar ± SD of viral titers for all biological replicates presented as individual data points. Two-way ANOVA analysis was used to determine statistically significant differences. The dashed line indicates the limit of detection for all graphs.

ulcerated and had evidence of macrophage and neutrophil accumulation within the airway lumen, whereas, H1N1pdm09-imm ferrets were similar to uninfected ferrets in that their bronchioles were clear of cellular debris with intact ciliated columnar lining epithelium (Supplementary Fig. 4). Furthermore, H1N1pdm09-imm alveolar interstitium had large airways that were clear with mild to moderate peripheral lymphocytic infiltrates and blood vessels that were multifocally surrounded by edema and lymphocytic infiltrates (Supplementary Fig. 4). In the absence of prior immunity, the large airways of the alveolar interstitium were partially ulcerated and filled with immune cells, and the alveolar spaces were filled with fibrin edema (Supplementary Fig. 4). These data indicate that pre-existing H1N1pdm09 immunity can reduce the pathology caused by α-swH1N2 infection.

Lastly, we examined the clinical outcomes for α-swH1N2 infected ferrets during these studies by cataloging the activity, weight loss, and other signs of the animals[47]. Intranasally α-swH1N2-infected ferrets with no prior immunity and H3N2-imm displayed a similar number of symptoms, while intranasally infected H1N1pdm09-imm ferrets displayed almost no symptoms (Fig. 8A). Ferrets with no prior immunity displayed a greater range of symptoms than those with pre-existing immunity (Fig. 8C). In airborne-infected animals, all ferrets, regardless of immunity, displayed a similar mean total number of symptoms (Fig. 8B), which included similar clinical signs over multiple days such as reduced activity scores, nasal discharge and weight loss (Fig. 8D). Overall, ferrets intranasally or airborne infected with α-swH1N2 had

mild symptoms, which varied by the category of symptoms. Taken together, although α-swH1N2 can still transmit between immune animals, its replication and severity of symptoms appear to be limited by prior immunity.

## Discussion

Identification of emerging respiratory viruses with pandemic potential is critical for enacting preparedness measures to mitigate their impact. Swine viruses are particularly concerning, given their agricultural importance that places them within close physical proximity to humans and the wide diversity of swine influenza strains[48]. Current risk assessment of pandemic threats is done through the WHO and CDC risk assessment tools[49,50], which use subject-area expert opinion to assign weighted scores for various categories and limited experimental data derived from multiple different in vitro and in vivo sources. In this study, we present a streamlined, adaptable strategy to experimentally triage influenza viruses that reduces the need for complete virus characterization since certain criteria must be met before proceeding to the next box in the decision tree. This pipeline represents a breathable framework that can and will be updated as additional data from characterization studies are conducted.

Using our decision tree, we analyzed representative circulating swine H1 strains from the alpha and gamma genetic clades that have a wide geographic distribution, are frequently detected in swine populations in the United States (Fig. 1C), and have exhibited sporadic human spillover events[51]. Previous representatives of the α-swH1N2

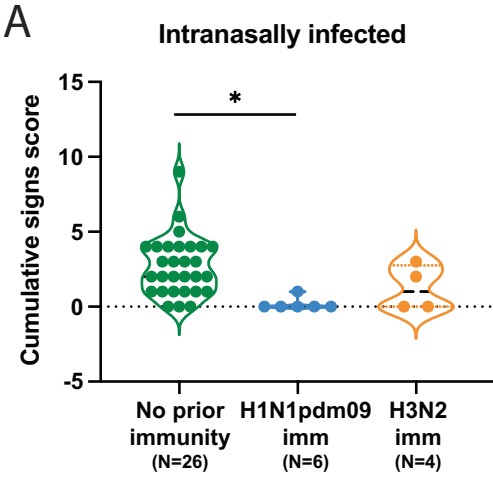

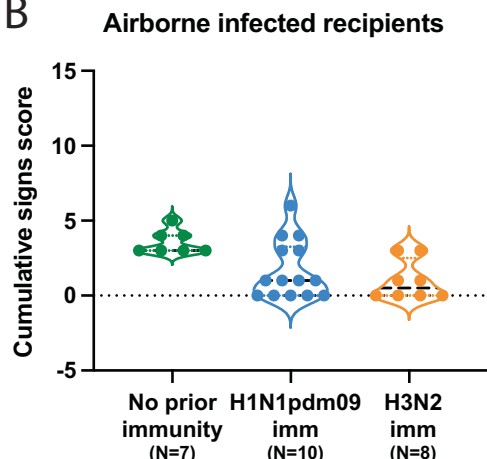

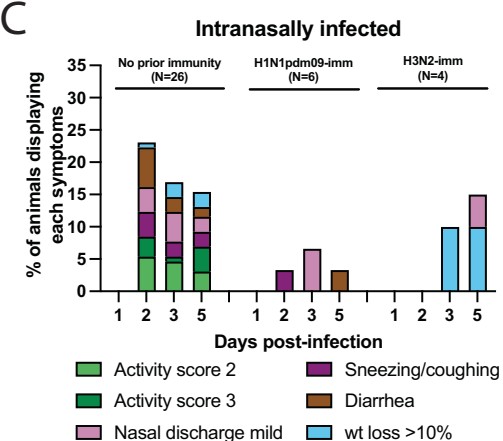

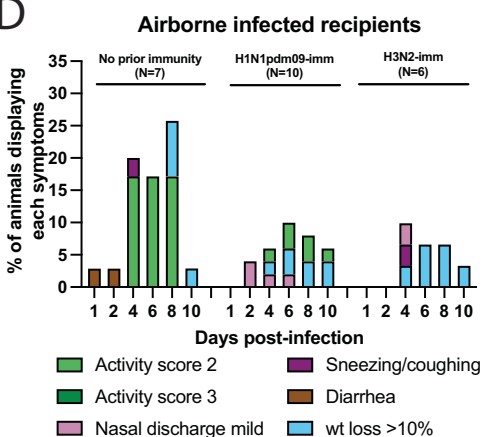

**Fig. 8 | Pre-existing H1N1pdm09 immunity reduces swine α-H1N2 influenza virus clinical signs. A** The symptoms for each intranasally α-swH1N2-infected ferret from Figs. 5 and 6 having either no prior immunity ($N = 22$), H1N1pdm09-imm ($N = 6$) or H3N2-imm ($N = 4$) were added together to assign each animal a cumulative score. Each dot represents the cumulative symptoms score for a single ferret. Two-way ANOVA analysis was used to determine statistically significant differences (*$p = 0.0452$). **B** The symptoms for each airborne α-swH1N2-infected ferret from Figs. 5 and 6 having either no prior immunity ($N = 7$), H1N1pdm09-imm ($N = 10$) or H3N2-imm ($N = 4$) were added together to assign each animal a cumulative score. Each dot represents the cumulative symptoms score for a single ferret. **C** Percent number of intranasally infected ferrets from panel A displaying each symptom on the indicated days post-infection. **D** Percent number of recipient ferrets from **B** displaying each symptom on the indicated days post-exposure. For **C** and **D**, wt = weight.

clade were shown to have antigenic distance from human vaccine strains, reduced recognition by human sera from two different cohorts[6], and transmitted efficiently from infected pigs to naive recipient ferrets[52]. While highly efficient at controlling antigenically similar influenza viruses, antibodies directed towards HA become less effective over each subsequent flu season as surface glycoproteins rapidly mutate through antigenic drift. No cross-neutralizing antibodies were detected against α-swH1N2 in H1N1pdm09- or H3N2-imm ferrets (Supplementary Table 2), suggesting that an initial infection with human seasonal viruses does not produce antibodies that cross-neutralize, and this was consistent with our human serum data (Fig. 3). Interestingly, human sera across all birth years tested had variable levels of anti-N2 antibodies (Supplementary Fig. 5), which may suggest that this NA-based immunity could provide some level of protection in a subset of the population[53–56]. Our prior work previously determined that prior immunity can influence the susceptibility to heterosubtypic viruses in a mechanism not mediated by neutralizing antibodies[40]. Thus, prior immunity from divergent strains can impact susceptibility of viruses through the air. We found that α-swH1N2 transmitted efficiently through the air to ferrets regardless of immune status, but the severity of disease after experimental infection with α-swH1N2 was

lower in animals with prior immunity. A similar phenomenon may explain the lower-than-expected morbidity and mortality of the 2009 pandemic in humans[57].

Protection against emerging influenza virus strains in hosts without neutralizing antibodies can be conferred from CD8+ T cells, which recognize conserved internal influenza virus proteins. Although prior adaptive immunity may not prevent influenza virus infection, CD8+ T cells that display cross-reactivity against different subtypes of influenza virus have been linked to more efficient clearance of virus and faster recovery from illness[58–60]. Indeed, prior immunity to human seasonal viruses was not protective against α-swH1N2 airborne infection (Fig. 5B, C). Encouragingly, experimentally infected ferrets with pre-existing immunity were able to clear α-swH1N2 faster and H1N1pdm09 immunity resulted in an overall decrease in virus shedding over time (Fig. 7B) and decreased lung pathology early during infection (Supplementary Fig. 4). However, the lack of disease severity in immune animals may also provide an opportunity for this virus to spread undetected and gain a foothold in the population, creating a pandemic risk. Taken together, our data demonstrate that this α-swH1N2 virus strain poses a higher pandemic risk than γ-swH1N1 that warrants continued surveillance efforts to capture zoonotic events and

an increased campaign to vaccinate swine against this H1 clade to reduce the amount of virus in source populations.

## Methods

### Genetic analysis and strain selection

All available swine H1 HA sequences from the USA collected between January 2019 and December 2021 were downloaded from the Bacterial and Viral Bioinformatics Research Center (BV-BRC)[61]. These data ($n = 2144$) were aligned with the World Health Organization (WHO)-recommended human seasonal H1 HA vaccine sequences and candidate vaccine sequences. The swine and human IAV HA sequences were aligned using mafft v7[62], and a maximum likelihood phylogeny for the alignment was inferred, following automatic model selection, using IQ-TREE v2[63] and visualized using smot v1.0.0[64] (Supplementary Fig. 1). The evolutionary lineage and genetic clade of each swine HA gene was identified using the BV-BRC Subspecies Classification tool, and the predominant clades and their geographic distribution were identified[4,65]. The 1 A and 1B lineages were detected in the USA and the genetic clades 1A.3.3.3 (38%, $n = 806$), 1B.2.1 (29%, $n = 622$), 1A.3.3.2 (12%, $n = 263$), 1A.1.1.3 (11%, $n = 233$), 1B.2.2.1 (5%, $n = 109$) and 1B.2.2.2 (3%, $n = 69$) represented 98% of detections. Given human variant detections, evidence for interspecies transmission, a significant reduction in cross-reactivity to human seasonal vaccines or candidate vaccine viruses, and limited detection by human population sera, we prioritized the 1A.1.1.3 and 1A.3.3.3 clades for characterization[7,21]. Representative selections within these clades were identified by generating an HA1 consensus sequence and identifying the best-matched field strain to the consensus that had an NA and internal gene constellation that reflected the predominant evolutionary lineages detected in surveillance (1A.1.1.3/α-swH1N2, A/swine/Texas/A02245420/2020, 97.2% to HA1 consensus: and 1A.3.3.3/γ-swH1N1, A/swine/Minnesota/A02245409/2020, 98.5% to consensus). Furthermore, the selected viruses had NA and internal gene patterns that matched the predominant evolutionary lineages detected between 2019–2021 (https://flu-crew.org/octoflushow/). The 1A.1.1.3/α-swH1N2 was paired with a N2-2002A gene with a TTTTPT internal gene constellation, and the 1A.3.3.3/γ-swH1N1 was paired with a N1-Classical gene with a TTTPPT internal gene constellation.

### Cells and viruses

Madin Darby canine kidney (MDCK) epithelial cells (ATCC, CCL-34) were maintained in Eagle's minimal essential medium supplemented with 10% fetal bovine serum, penicillin/streptomycin and L-glutamine. Primary HBE cell cultures were derived from human lung tissue that were differentiated and cultured at an air-liquid interface using a protocol approved by the relevant institutional review board at the University of Pittsburgh[66]. The influenza A virus strains, A/swine/Texas/A02245420/2020 (α-swH1N2, 1A.1.1.3) and A/swine/Minnesota/A02245409/2020 (γ-swH1N1, 1A.3.3.3) were obtained from the National Veterinary Services Laboratories (NVSL) repository for the USDA IAV-S surveillance system. Reverse genetic derived strains of A/California/07/2009 (H1N1pdm09) and A/Perth/16/2009 (H3N2) were a generous gift from Dr Jesse Bloom (Fred Hutch Cancer Research Center, Seattle). The eight plasmids were transfected using Mirus TransIT-LT1 transfection reagent into 293 T cells (ATCC). Supernatants were collected at 72-h post-transfection and used to infect a monolayer of MDCK cells to produce a cell passage stock of virus.

### TCID50 assay

MDCK cells were seeded at a density of 10,000 cells per well in 96-well plate three days prior to the assay. Cells were washed with sterile phosphate-buffered saline (PBS) followed by addition of 180 μL of Eagle's minimal essential medium supplemented with Anti-Anti, L-glutamine and 0.5 μg/mL TPCK-treated trypsin. 20 μL of virus was diluted in the first row and tenfold serial dilutions on cells were performed. The assay was carried out across the plate with the last row as the cell control without virus. The cells were incubated for 96 h at 37 °C in 5% CO2 and scored for cytopathic effect (CPE).

### Serological assays

Hemagglutination inhibition (HAI) was used to assess the presence of receptor-binding antibodies to HA protein from the selected viruses in human sera. Briefly, one-part sera were treated with three parts receptor destroying enzyme (RDE) overnight at 37°C to remove non-specific inhibitors. The following day, the sera was heat inactivated at 56 °C for 30 min and six parts of normal saline added. In a V-bottom microtiter plate, two-fold serial dilutions of RDE-treated sera were performed and incubated with eight hemagglutinating units of virus for 15 min. Turkey red blood cells were added at a concentration of 0.5% and incubated for 30 min. The reciprocal of the highest dilution of serum that inhibited hemagglutination was determined to be the HAI titer. The titer of neutralizing antibodies was determined using the microneutralization assay. Human or ferret sera was heat inactivated at 56 °C for 30 min and serially diluted 2-fold in a 96-well flat-bottom plate. $10^{3.3}$ TCID$_{50}$ of influenza virus was incubated with the sera for 1 h at room temperature before being transferred to a 96-well plate on confluent MDCK cells. Sera was maintained for the duration of the experiment and CPE was determined on day 4 post-infection. The neutralizing titer was expressed as the reciprocal of the highest dilution of serum required to completely neutralize the infectivity of $10^{3.3}$ TCID$_{50}$ of virus on MDCK cells. The concentration of antibody required to neutralize 100 TCID$_{50}$ of virus was calculated based on the neutralizing titer dilution divided by the initial dilution factor, multiplied by the antibody concentration.

### Glycan array

Glycan arrays were prepared as previously described[33,67]. Briefly, glycans were prepared at 100 μM in 150 mM Na3PO4 buffer (pH 8.4) and printed onto NHS-activated glass microscope slides (SlideH, Schott) using a MicroGridII (Digilab) contact microarray printer equipped with Stealth Microarray Pins (SMP3, ArrayIt). Residual NHS was blocked by treatment with 50 mM ethanolamine in 50 mM borate buffer, pH 9.2 for 1 h and washed with water. Slides were centrifuged to remove excess water and were stored at −20C. For analysis of receptor specificity glycan arrays were overlayed with culture fluid containing intact influenza virus prepared in MDCK cells for one hour at room temperature. Slides were then washed with phosphate buffered saline (PBS) and water, followed by incubation with biotinylated Galananthus Novalis Lectin (GNL; Vector Labs) at 1ug/mL in 1X PBS for 1 h[67]. Sides were washed with PBS and overlayed with 1 μg/ml Streptavidin-AlexaFluor488 (LifeTech) for 1 h and washed with PBS and water. Slides were then scanned using an Innoscan 1100AL microarray scanner (Innopsys). Signal values are calculated from the mean intensities of 4 of 6 replicate spots with the highest and lowest signal omitted and graphed.

### Replication kinetics

Four different HBE patient cell cultures were used (HBE0344, HBE0338, HBE0342, HBE0370). The apical surface of the HBE cells was washed in PBS and $10^3$ TCID$_{50}$ of virus was added per 100 μL of HBE growth medium. After 1 hour incubation, the inoculum was removed and the apical surface was washed three times with PBS. At the indicated time points, 150 μL of HBE medium was added to the apical surface for 10 min to capture released virus particles. The experiment was performed in triplicate in at least three different patient cell cultures. Infectious virus was quantified by TCID$_{50}$ using the endpoint method[68], as described above.

### Enzyme-linked lectin assay (ELLA)

The neuraminidase activity was determined using a peanut-agglutinin based ELLA. A 96-well ultra-high binding polystyrene plate was coated

with 25 μg/mL of fetuin diluted in coating buffer overnight at 4 °C and the excess fetuin was removed using wash buffer (0.01 M PBS, pH 7.4, 0.05% Tween 20). Two-fold serial dilutions of $10^{7.5}$ TCID$_{50}$/mL virus stock or 62.5 mU/mL *Clostridium perfringes* neuraminidase (to standardize the viruses between different plates) were performed in a 96-well plate. Serial dilutions were then transferred to the plates coated with fetuin and incubated overnight at 37 °C. Plates were thoroughly washed 6 times with wash buffer and incubated in the dark at room temperature with peroxidase-labeled peanut agglutinin solution for 2 h. O-phenylenediamine dihydrochloride substrate was added for 10 min to and the reaction was stopped using sulfuric acid. Absorbance was read at 490 nm. NA activity was assayed in duplicate and performed in three independent replicates.

#### pH inactivation assay
The pH of inactivation assay[69] was used to determine the pH at which HA undergoes its irreversible conformational change. 10 μL of virus stock was incubated in 990 μL of PBS adjusted to the indicated pH values for 1 h at 37 °C and immediately neutralized by titering on MDCK cells using the TCID$_{50}$ endpoint titration method[68] to determine the remaining infectious virus titer. The pH that reduced the titer by 50% (EC$_{50}$) was calculated by regression analysis of the dose-response curves. Each experiment was performed in triplicate in at least three independent biological replicates.

#### Stability of stationary droplets
Desiccator chambers containing saturated salt solutions of potassium acetate, potassium carbonate, magnesium nitrate or sodium chloride were equilibrated to 23%, 43%, 55% of 75% relative humidity (RH), respectively. Ten 1 μL droplets of HBE-propagated virus were spotted onto a 6-well plate in duplicate and immediately incubated in the desiccator chamber for 2 h. Chambers were maintained in a biosafety cabinet for the duration of the experiment and a HOBO UX100011 data logger was used to collect RH and temperature data. After 2 h, the droplets were collected in 500 μL of L-15 medium, which was titered on MDCK cells using the TCID$_{50}$ endpoint method[68]. Decay was determined by subtracting the titer of the virus aged for 2 hours from the titer of the virus that had been deposited and then immediately recovered.

#### Animal ethics statement
Ferret experiments were conducted in a BSL2 facility at the University of Pittsburgh in compliance with the guidelines of the Institutional Animal Care and Use Committee (approved protocol 22061230). Animals were sedated with isoflurane following approved methods for all nasal washes and survival blood draws. Ketamine and xylazine were used for sedation for all terminal procedures, followed by cardiac administration of euthanasia solution. Approved University of Pittsburgh Division of Laboratory Animal Resources (DLAR) staff administered euthanasia at time of sacrifice.

#### Human subjects research ethics statement
Human serum samples used in this study were collected from healthy adult donors who provided written informed consent for their samples to be used in infectious disease research. Participants responded to a notification about the study from flyers and/or website announcements, so a self-selection bias is possible if individuals who elected to participate in research were more likely to receive their annual influenza vaccinations. This should not have introduced bias with respect to community influenza exposures based upon age. The University of Pittsburgh Institutional Review Board approved this protocol (STUDY20030228). All participants were consented by trained staff and self-reported their age, sex, race, ethnicity, residential zip code, history of travel and immunization. HBE cultures are obtained from deidentified patients under an approved protocol from The University of Pittsburgh Institutional Review Board (STUDY19100326) and provided from the tissue airway core to for these studies.

#### Ferret screening
Four- to six-month-old male ferrets were purchased from Triple F Farms (Sayre, PA, USA). All ferrets were screened by HAI for antibodies against circulating influenza A and B viruses, as described in 'Serology' section. The following antigens were obtained through the International Reagent Resource, Influenza Division, WHO Collaborating Center for Surveillance, Epidemiology and Control of Influenza, Centers for Disease Control and Prevention, Atlanta, GA, USA: 2018–2019 WHO Antigen, Influenza A (H3) Control Antigen (A/Singapore/INFIMH-16-0019/2016), BPL-Inactivated, FR-1606; 2014–2015 WHO Antigen, Influenza A (H1N1)pdm09 Control Antigen (A/California/07/2009 NYMC X-179A), BPL-Inactivated, FR-1184; 2018–2019 WHO Antigen, Influenza B Control Antigen, Victoria Lineage (B/Colorado/06/2017), BPL-Inactivated, FR-1607; 2015–2016 WHO Antigen, Influenza B Control Antigen, Yamagata Lineage (B/Phuket/3073/2013), BPL-Inactivated, FR-1403.

#### Ferret infections
To generate ferrets with pre-existing immunity against seasonal influenza viruses, ferrets were inoculated intranasally with $10^6$ TCID$_{50}$ in 500 μL (250 μL in each nostril) of recombinant A/California/07/2009 or A/Perth/16/2009. These animals were allowed to recover and housed for 126 to 137 days before acting as a recipient in a transmission experiment or being similarly infected with A/swine/Texas/A02245420/2020.

#### Transmission studies
The transmission cage setup was a modified Allentown ferret and rabbit bioisolator cage[12,32]. For each study, four ferrets were anesthetized with isoflurane and inoculated intranasally with $10^6$ TCID$_{50}$ in 500 μL (250 μL in each nostril) of A/swine/Texas/A02245420/2020 to act as donors. Twenty-four hours later, a naïve or immune recipient ferret was placed into the adjacent cage, which is separated by two staggered perforated steel plates welded together one inch apart with directional airflow from the donor to the recipient. Recipients were exposed to the donors for 2 days with nasal washes being collected from each donor and recipient every other day for 11 days. For the transmission chain experiment (Fig. 6), after the initial 2-day exposure, the recipients were transferred to the donor side of a new transmission cage where a naïve recipient ferret was on the other side of the divider. These animals were subsequently singled housed following 48 h. To prevent accidental contact or fomite transmission by investigators, the recipients were handled first and extensive cleaning of gloves, sedation chamber, biosafety cabinet, and temperature monitoring wand was performed between each pair of animals. Sera from donor and recipient ferrets were collected upon completion of experiments to confirm seroconversion. To ensure no accidental contact or fomite transmission during husbandry procedures, recipient animal sections of the cage were cleaned prior to the donor sides, with one cage being done at a time. Fresh scrapers, gloves, and sleeve covers were used for each subsequent cage cleaning. Clinical symptoms such as weight loss and temperature were recorded during each nasal wash procedure and other symptoms such as sneezing, coughing, activity, diarrhea or nasal discharge were noted during any handling events. Once animals reached 10% weight loss, their feed was supplemented with A/D diet twice a day to entice eating. Clinical scoring was previously described in ref. 47. Briefly, disease signs were noted for each infected ferrets every day that a procedure was performed and included activity score 2 (alert but not playful), activity score 3 (neither alert nor playful), mild nasal discharge, sneezing or coughing, weight loss (10% to 15%) or diarrhea. The total number of disease signs was added together for each intranasally or airborne infected ferret over the course of the study to provide the cumulative signs score.

## Tissue collection and processing

The respiratory tissues were collected from euthanized ferrets aseptically in the following order: entire right middle lung, left cranial lung (a portion equivalent to the right middle lung lobe), one inch of trachea cut lengthwise, entire soft palate, and nasal turbinates. Tissue samples were weighed, and Leibovitz's L-15 medium was added to make a 10% (lungs) or 5% (trachea) w/v homogenate. The soft palate and nasal turbinates were homogenized in 1 mL of Leibovitz's L-15 medium. Tissues were dissociated using an OMNI GLH homogenizer (OMNI International) and cell debris was removed by centrifugation at $900 \times g$ for 10 min. Influenza virus titers were determined by endpoint $TCID_{50}$ assay[68] as described above. The lungs were fixed in 10% neutral buffered formalin and subsequently processed in alcohols for dehydration and embedded in paraffin wax. Sections were stained with haematoxylin and eosin (H&E). The sections were examined 'blind' to experimental groups to eliminate observer bias by a board-certified animal pathologist (LHR).

## Data availability

The source data generated, analyzed, and presented in Figs. 1, 3–8, Supplementary Figs. 3 and 5 of this current study have been archived on FigShare (https://doi.org/10.6084/m9.figshare.24926505.v2). Source data for Fig. 3 on FigShare includes sex data on the various human samples analyzed in Fig. 3. Phylogenetic analyses, associated strain selection data and accession identifiers are archived at https://github.com/flu-crew/datasets/tree/main/h1n2-pandemic_risk.

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

## Acknowledgements

This project has been funded in part with Federal funds from the National Institute of Allergy and Infectious Diseases, National Institutes of Health, Department of Health and Human Services, under Contract No. 75N93021C00015; the United States Department of Agriculture, Agricultural Research Service, project number 5030-32000-231-000-D, Cystic Fibrosis Foundation Research Development Program to the University of Pittsburgh, and Burroughs Wellcome CAMS 1013362.02 to A.K.M. We thank Dr. Daniel Perez for generously providing plasmids. We thank Dr. Rachel Duron for critical review and feedback. We thank members of the Lakdawala lab for critical feedback and the University of Pittsburgh DLAR staff for assistance with animal studies. The funders had no role in study design, data collection and interpretation, or the decision to submit the work for publication. Mention of trade names or commercial products in this article is solely for the purpose of providing specific information and does not imply recommendation or endorsement by the USDA. USDA is an equal opportunity provider and employer.

## Author contributions

V.L. and S.S.L. designed the experiments, analyzed, interpreted the data and wrote the manuscript. V.L., N.C.R., K.R.M., A.J.F., M.J.S., R.M., J.E.J., S.G.W. and L.H.R. performed the experiments. J.D.D., L.X., D.J.B., S.W.,

S.A.F., M.M.M., J.C.P., A.K.M., T.K.A. and A.L.V.B. contributed resources and analysis. All authors edited and approved the manuscript.

## Competing interests

The authors declare no competing interests.
