## [Peer Review File · Nature Communications]

Potential pandemic risk of circulating swine H1N2 influenza virusesEditorial Note: This manuscript has been previously reviewed at another journal. This document only contains reviewer comments and rebuttal letters for versions considered at Nature Communications.

REVIEWERS' COMMENTS

Reviewer #1 (Remarks to the Author):

All my concerns are appropriately addressed. Thank you.

Reviewer #2 (Remarks to the Author):

The authors have addressed most points well and have performed experiments that demonstrate that H1N2 transmission is inefficient in experiments involving transmission chains in partially H1N1 immune animals (50-33%). Furthermore, transmission is not very efficient to H3N2 immune recipients (50%-100%). Humans would have both H1N1 and H3N2 immunity. Nevertheless the authors stress the pandemic potential of this virus to which the population in the US has been exposed for years without more than a handful of infections.

Reviewer #3 (Remarks to the Author):

The revised manuscript and the author's responses have adequately addressed each of my comments. I have no further comments.

Response to Reviewers:

Potential pandemic risk of circulating swine H1N2 influenza viruses

Le Sage, et al.

Reviewer comments are in black italicized font and author responses are in **normal bold font**. Referenced line numbers correspond to the revised version of the manuscript *without* tracked changes.

REVIEWERS' COMMENTS

Reviewer #1 (Remarks to the Author):

All my concerns are appropriately addressed. Thank you.

Reviewer #2 (Remarks to the Author):

The authors have addressed most points well and have performed experiments that demonstrate that H1N2 transmission is inefficient in experiments involving transmission chains in partially H1N1 immune animals (50-33%). Furthermore, transmission is not very efficient to H3N2 immune recipients (50%-100%). Humans would have both H1N1 and H3N2 immunity. Nevertheless the authors stress the pandemic potential of this virus to which the population in the US has been exposed for years without more than a handful of infections.

We thank the reviewer for this comment and agree that prior immunity does potentially reduce the impact of the α -swH1N2 strain, as shown by our data in this manuscript. We highlight this idea throughout the manuscript in the abstract (lines 39-41), results (lines 288-290), and discussion (lines 320-322). However, since transmission can still occur, even in the absence of severe symptoms, this provides an opportunity for the α -swH1N2 strain to continue circulating and change further, creating a greater pandemic risk. We have revised our discussion to highlight that the α -swH1N2 strain is a greater pandemic risk than the γ -swH1N1 strain because of this and that continued surveillance is warranted (lines 334-339).

Reviewer #3 (Remarks to the Author):

The revised manuscript and the author's responses have adequately addressed each of my comments. I have no further comments.